# Structure-guided identification of mitogen-activated protein kinase-1 inhibitors towards anticancer therapeutics

**Md Nayab Sulaimani**[1], **Shazia Ahmed**[2], **Farah Anjum**[3], **Taj Mohammad**[1], **Anas Shamsi**[4]\*, **Ravins Dohare**[1]\*, **Md. Imtaiyaz Hassan**[1]\*

**1** Centre for Interdisciplinary Research in Basic Sciences, Jamia Millia Islamia, New Delhi, India, **2** Department of Computer Science, Jamia Millia Islamia, New Delhi, India, **3** Department of Clinical Laboratory Sciences, College of Applied Medical Sciences, Taif University, Taif, Saudi Arabia, **4** Center of Medical and Bio-Allied Health Sciences Research (CMBHSR), Ajman University, Ajman, United Arab Emirates

\* anas.shamsi18@gmail.com (AS); ravins@jmi.ac.in (RD); mihassan@jmi.ac.in (MIH)

**Data Availability Statement:** All relevant data are within the manuscript and its Supporting Information files.

## Abstract

Mitogen-activated protein kinase 1 (MAPK1) is a serine/threonine kinase that plays a crucial role in the MAP kinase signaling transduction pathway. This pathway plays a crucial role in various cellular processes, including cell proliferation, differentiation, adhesion, migration, and survival. Besides, many chemotherapeutic drugs targeting the MAPK pathway are used in clinical practice, and novel inhibitors of MAPK1 with improved specificity and efficacy are required. Hence, targeting MAPK1 can be crucial to control metastasis in cancer therapeutics. In this study, we utilized a structure-guided virtual screening approach to screen a library of thousands of natural compounds from the ZINC database. The Lipinski rule of five (RO5) was used as a criterion for the primary selection of natural compounds. The screened compounds were prioritized based on their binding affinity, docking scores, and specificity towards the kinase domain of MAPK1 during the molecular docking process. Subsequently, the selected hits underwent rigorous screening that included the identification of potential pan-assay interference compounds (PAINS), ADMET evaluation, and prediction of pharmacological activities using PASS analysis. Afterwards, we performed a comprehensive interaction analysis to explore the binding prototypes of the screened molecules with the key residues within the MAPK1 kinase domain. Finally, selected molecules underwent extensive all-atom molecular dynamics (MD) simulations for a time duration of 200 nanoseconds. The study pinpointed three natural compounds with ZINC database IDs ZINC0209285, ZINC02130647, and ZINC02133691 as potential inhibitors of MAPK1. The study highlights that these compounds could be explored further in preclinical and clinical investigations to develop anticancer therapeutics.

## 1. Introduction

In humans, the *MAPK1* gene encodes the protein mitogen-activated protein kinase 1 (MAPK 1), often referred to as ERK2. Ras/Raf/MEK/ERK is the key signaling pathway cascade that

**Funding:** This work is supported by Council of Scientific and Industrial Research, Government of India (Grant No. 27/0368/2020). This research was funded by Taif University, Saudi Arabia, Project Number (TU-DSPP-2024-140). The funders had no role in study design, data collection and analysis, decision to publish, or preparation of the manuscript.

**Competing interests:** The authors have declared that no competing interests exist.

combines external clues from cell surface receptors with gene expression and protein regulation from several cellular components [1]. It is a highly conserved serine-threonine kinase that is involved in various biological processes. It can independently transduce extracellular signals into the cells to control the expression of related genes [2]. Therefore, it is essential for many physiological events, including metabolism, development, memory formation, and immunity. MAPK1 is the subfamily of the MAPK pathway. It has abnormal expression in various cancerous diseases, as suggested by previous research [3]. Additionally, it has been noted that HeLa cell interference with MAPK1 expression can dramatically reduce cancer cell growth and trigger cell death. According to certain studies, cervical cancer cells' ability to undergo the epithelial-mesenchymal transition (EMT) may be strongly aided by the activation of the MAPK1 signalling pathway [4].

One of the biggest problems with public health is cancer. It is a complicated disease with a wide range of etiologies, symptoms, and prognoses. Each year, millions of people are affected by the various cancers worldwide. According to the assumption by the International Agency for Research on Cancer (IARC), there would be 16.4 million deaths associated with cancer-related disease and 29.5 million new cases could be noticed globally in 2040, an increase of roughly 1.6–1.7 times from 2018 predictions [5]. The scientific and pharmaceutical communities have focused a great deal of emphasis on protein kinases in the past ten years to develop small molecule inhibitors. MAPK1 is a dual-specificity kinase that needs both serine and threonine residues to be phosphorylated to become catalytically active. MAPK1 is a member of the class of dual-specificity kinases [6]. MAPK1 creates a point of convergence for numerous upstream pathways, providing a powerful inhibitory effect. The identification of MAPK1 inhibitors may have implications for several diseases, most notably cancer progression. Here, we implicated a structure-based rational drug design approach for targeting MAPK1 in order to find new leads that may have a potent inhibitory effect [7]. The structural organization of MAPK1 is shown in Fig 1.

Nowadays, drug development is a multifaceted process involving academia, industry, and innovative approaches. Academic institutions contribute significantly, with success rates

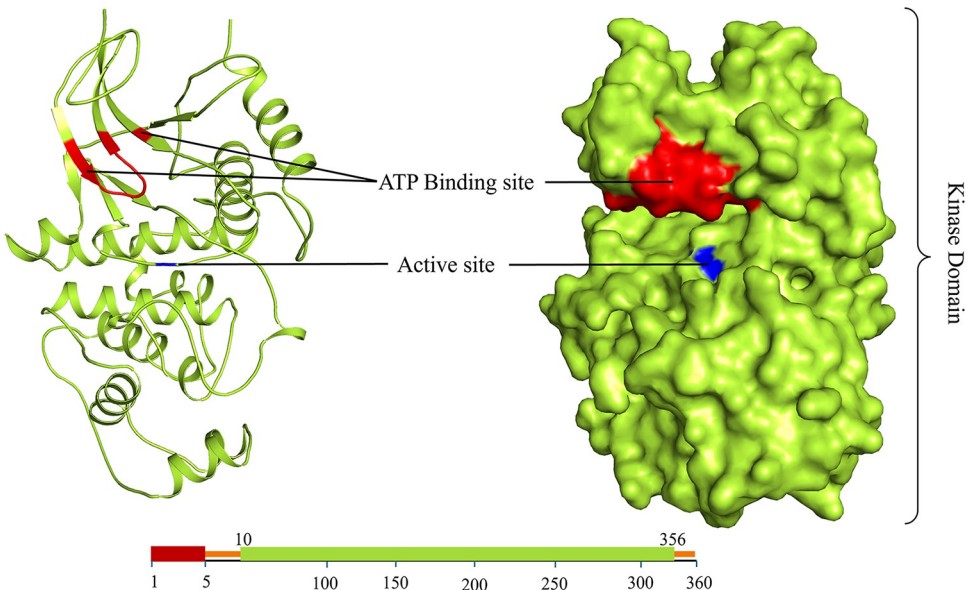

**Fig 1. Cartoon and surface representation of MAPK1 protein structure highlighting ATP binding site and active site indicating key functional regions critical for its biological activity.**

varying across phases (Phase I: 75%, Phase II: 50%, Phase III: 59%, NDA (New Drug Application)/ BLA (Biologics License Application) phase: 88%). Challenges persist, but innovations like artificial intelligence and in vitro technologies aim to accelerate research and development. Drug repurposing and molecular-level disease understanding also play key roles. The future lies in personalized, effective, and non-toxic drug design [8].

In this study, we screened about ~22000 natural compounds that were taken from the ZINC database. A molecular docking-based virtual screening approach was utilized to get the high-affinity molecules that can bind with the MAPK1. We conducted ADMET analysis, followed by PASS (Prediction of Activity Spectra for Substances) analysis, on the chosen compounds. Subsequently, we screened the compounds based on their specific interactions with the MAPK1 binding pocket by utilizing the top hits generated. Furthermore, for deeper insights into the dynamic behavior of the protein and its complex, all-atom MD simulations were performed for the time duration of 200 ns [9].

In recent years, a few MAPK1 inhibitors have been developed and are under clinical evaluation, as suggested by previous studies [10]. Multiple methods have recently been combined to develop diverse classes of MAPK1 inhibitors to control different types of cancer [11]. However, the MAPK1 inhibitors currently on the market are not specific and can have both on- and off-target effects. Therefore, safe and effective MAPK1 inhibitors must be developed for the treatment of cancer and its related disorders. Our approach of using the ZINC database for structure-based virtual screening for the discovery of potent MAPK1 inhibitors for the development of therapeutic compounds for the treatment of cancer. After the necessary changes, the identified molecule with better pharmacological characteristics can be further assessed for the development of selective MAPK1 inhibitors [12].

## 2. Material and methods

### 2.1. Computational resources

A systematic approach of structure-guided virtual screening based on molecular docking and simulation studies was carried out using various bioinformatics software, such as InstaDock [13], Discovery Studio Visualizer [14], GROMACS [15], and PyMOL [16]. The RCSB-Protein Data Bank (PDB), pkCSM server, QtGrace, SigmaPlot, SwissADME, and other tools were utilized for data interpretation and retrieval. The 3-dimensional structure of the protein was retrieved from the RCSB Protein Data Bank (https://www.rcsb.org/) with PDB ID: 8AOJ at the resolution of 1.12 Å. We have used PyMOL to process the structure by isolating the kinase domain from all coordinates and examining the structure and heteroatoms. Water molecules and co-crystallized ligands were removed from the initial coordinates. The structure of MAPK1 was further refined by remodelling all the missing residues, followed by adding the hydrogen atom to the polar group, and then the appropriate atom types were assigned for the docking-based screening in InstaDock. The library containing ~22000 RO5 filtered compounds was taken from the ZINC database for screening [15, 17].

### 2.2. Molecular docking-based virtual screening

Molecular docking is a computational technique that determines a most suitable orientation of a molecule while binding to its receptor. It is used to see interactions of ligands with the receptor, typically proteins [18]. In this study, docking was carried out to obtain the binding affinity and the best conformational pose of the selected compounds for the MAPK1. Molecular docking was performed on the high-performance workstation with Intel i7, the 14th generation, with 28 CPU cores and a Windows 10 operating system. In this investigation, protein-ligand docking was conducted using InstaDock with blind search space with a grid size of 75 Å, 60 Å,

and 80 Å for X, Y, and Z coordinates, respectively. The center of the grid was set to X: −1.85, Y: 5.23, Z: 38.32 axes. Furthermore, based on the ligand efficiency and energy values, the docking result was filtered. Discovery Studio Visualizer and PyMOL were used to visualize the structures for the investigation of bound conformation and various interactions between MAPK1 and selected compounds.

## 2.3. Pharmacokinetic evaluation

It was not expected that the annual increase in newly developed structures would be accompanied by a rise in the number of drugs that are commercialized in the market. Poor pharmacokinetic characteristics of the compounds have been a reason for this [19]. Thus, appropriate screening filters for ADMET parameters are crucial. In this study, the compounds obtained from docking were analyzed based on their physicochemical and ADMET properties using the SwissADME [20] and pkCSM web servers [21].

## 2.4. Interaction analysis

The exploration of 2D interactions offers valuable insights into residual interactions, inhibitory patterns, and bond types between receptors and ligands. This analysis was facilitated using Discovery Studio Visualizer, followed by validation using PyMOL to confirm the binding and interaction patterns of three selected compounds identified through PASS analysis [22]. The resulting out files of these three compounds revealed a total of 27 docked conformations, which were evaluated by analyzing interacting residues as an initial step. Hydrogen bonding and other interactions, such as pi-pi bonds, are essential for drug development because they improve the structural stability, catalysis, and biological activity characterization of receptors on ligand binding [23]. Hydrogen bonds to catalytically important residues can stabilize the transition state of the enzymatic reaction, thereby lowering the activation energy required for the reaction to proceed. This stabilization is crucial for enhancing the efficiency of the interaction. These hydrogen bonds often result in stronger interactions between the ligands and the target protein, leading to higher binding affinity. This is because the catalytically important residues are typically located in key positions within the active site, where they can form optimal interactions with the ligand. Also, they can contribute to the selectivity of the ligand for the target protein. The angle and distance cut-off for hydrogen bonds between donor (D) and acceptor (A) were set to 3.5 Å and 150–180˚, respectively. These hydrogen bonds can help maintain the proper conformation of the enzyme's active site, ensuring that it remains in an optimal state for the interaction.

## 2.5. Molecular dynamics simulations

The virtual screening discoveries led to the molecular mechanics level, where we conducted a 200 ns MD simulation on three selected compounds along with free state MAPK1 at 300 K. Simulation were done using GROMACS software employing the Chemistry at Harvard Macromolecular Mechanics (CHARMM) forcefield [24]. For each compound, topologies and force field parameters were developed by using CGenFF (https://cgenff.com/). Each ligand-MAPK1 system was simulated in a virtual cubic box of water with a dimension of 10 Å and solvated by using *gmx solvate* module in the TIP3P water model. The TIP3P model is relatively simple, consisting of three interaction sites, which makes it computationally efficient. TIP3P is very compatible with the CHARMM force field, making it a better choice for MD simulation.

 All systems were energy-minimized using the steepest descent algorithm followed by charge neutralization. The temperature of all systems was gradually heated from 0 to 300 K over a 1000 ps equilibration phase at constant volume under periodic boundary conditions. The final

MD run of 200 ns was performed on all the systems. Quality check metrics, such as kinetic energy, volume, density, and enthalpy, were used to authenticate the obtained simulations for MAPK1 and its ligand complexes [18]. The simulated GROMACS trajectories generated on parameters including RMSD (root mean square deviation), RMSF (root mean square fluctuation), $Rg$ (radius of gyration), SASA (solvent accessible surface area), PCA (principle component analysis), and others, for each residue with respect to time function, wherein QtGrace plotted graphs to illustrate MAPK1 residual interaction and stability on ligand binding [25].

## 3. Results and discussion

### 3.1. Molecular docking-based virtual screening

A library of ~22000 compounds from the ZINC database was utilized for a comprehensive strategy of virtual screening to discover MAPK1 selective inhibitors. After the docking run, the docking software created log files and out files for each compound, which included affinity scores and docked poses, respectively. It helps eliminate compounds based on unsuitable binding affinities, docking score, and binding position [26]. The virtual screening suggested a significant number of promising hits with a high binding affinity towards the MAPK1 binding cavity, making them suitable candidates for additional screening for MAPK1 inhibitors [27]. The filtering of docked output revealed 100 hits out of 22000 compounds with a significant binding affinity to MAPK1 ranging from −12.0 to −10.3 kcal/mol (**S1 Table**). Since a higher binding affinity value indicates a more stable receptor-ligand complex, we anticipate increased stability in the complexes created using these 100 selected compounds [28]. **Table 1** shows the binding affinity of the top 10 compounds selected based on their physicochemical and pharmacokinetics properties and a control molecule. Structures of these compounds are shown in S2 Table. The docking results revealed that the chosen compounds have a great ability to bind MAPK1 with substantial ligand efficiency. These findings suggested that the molecules that were chosen could be turned into novel MAPK1 inhibitors for therapeutic development in cancer treatment.

### 3.2. Physicochemical and pharmacokinetic properties

To evaluate the ADMET profile of the top ten selected compounds and a control molecule, Ulixertinib, we used the pkCSM online server. The SMILE strings of all structures were obtained from Discovery Studio Visualizer and used as input to predict ADMET properties [29]. A ligand's physicochemical and pharmacokinetic characteristics may determine whether

**Table 1. Selected hits and a control molecule with their docking scores with MAPK1.**

| S. No. | Ligand | Binding free energy (kcal/mol) | pKi | Ligand efficiency (kcal/mol/non-H atom) |
|---|---|---|---|---|
| 1. | ZINC02161110 | −11.6 | 8.51 | 0.3053 |
| 2. | ZINC02161108 | −11.2 | 8.21 | 0.2947 |
| 3. | ZINC03844856 | −11.1 | 8.14 | 0.3083 |
| 4. | ZINC02092851 | −11.0 | 8.07 | 0.3056 |
| 5. | ZINC02161106 | −10.9 | 7.99 | 0.2868 |
| 6. | ZINC02119958 | −10.8 | 7.92 | 0.3375 |
| 7. | ZINC04083885 | −10.7 | 7.85 | 0.3147 |
| 8. | ZINC03839446 | −10.6 | 7.77 | 0.2865 |
| 9. | ZINC02130647 | −10.5 | 7.70 | 0.31 |
| 10. | ZINC02133691 | −10.4 | 7.63 | 0.3059 |
| 11. | Ulixertinib (control) | −9.2 | 6.75 | 0.3172 |

**Table 2. ADMET properties of the 10 selected natural compounds and a control molecule.**

| S. No. | Compound ID | Absorption (GI absorption) | Distribution (BBB permeability) | Metabolism (CYP2D6 inhibitor) | Excretion (OCT2 Substrate) | Toxicity (AMES/Hepatoxic) |
|---|---|---|---|---|---|---|
| 1. | ZINC02161110 | 94.81 | −0.25 | No | No | No |
| 2. | ZINC02161108 | 94.81 | −0.25 | No | No | No |
| 3. | ZINC03844856 | 100 | −1.02 | No | No | No |
| 4. | ZINC02092851 | 100 | −0.97 | No | No | No |
| 5. | ZINC02161106 | 94.81 | −0.25 | No | No | No |
| 6. | ZINC02119958 | 100 | −0.55 | No | No | No |
| 7. | ZINC04083885 | 95.77 | −0.12 | No | No | No |
| 8. | ZINC03839446 | 91.19 | −1.3 | No | No | No |
| 9. | ZINC02130647 | 98.95 | −0.83 | No | No | No |
| 10. | ZINC02133691 | 96.36 | −0.95 | No | No | No |
| 11. | Ulixertinib (control) | 88.28 | −0.84 | No | No | Yes |

it would make a good drug candidate and how likely it would succeed in clinical trials. The results of the finding show that every molecule met Lipinski's rule of five, which is a key formula for determining a drug's likelihood [30]. To assess the ADMET characteristics and PAINS filter of the chosen hit compounds from the docking study, we used two tools, pkCSM and SwissADME [31].

The top 10 selected ligands exhibit the highest binding affinity and specificity towards the active site of MAPK1. **Table 2** shows that these compounds have been selected for their favourable ADMET properties and lack PAINS features. All the compounds have shown significant binding affinity, surpassing that of the known control, Ulixertinib, indicating these are making a more stable complex with MAPK1. Also, Ulixertinib was found to be hepatotoxic after an ADMET analysis. Out of 10, the top three compounds were prioritized based on their biological activities like antineoplastic, kinase inhibitory and anti-inflammatory activities with considerably high Pa values. Overall, the results demonstrate that the ten compounds have favorable physicochemical characteristics without any PAINS patterns, indicating that they may be effective leads for drug development [32].

## 3.3. PASS analysis

PASS state that compounds with $P_a > P_i$ is considered to be the desired ones to show specific biological activity. The $P_a$ (probability to be "Active") value indicates the probability that the compound will exhibit a specific biological activity. Whereas $P_i$ (probability to be "Inactive") indicates the probability of the compound not showing any specific activity [33]. PASS predicts multiple biological activity types simultaneously based on chemical compound structures. Before chemical synthesis and biological testing, PASS prediction is a helpful method for forecasting the biological activity profiles of the molecules. The Way2Drug online server performed a PASS analysis on all selected compounds to predict biological activity, i.e., anti-cancer activities, to provide potential therapeutic leads for MAPK1-mediated cancer [34]. PASS analysis results showed that the three selected compounds, ZINC02092851, ZINC02130647, and ZINC02133691, showed favorable biological properties (Table 3). These compounds may act as an anti-inflammatory, antineoplastic, kinase inhibitor, phosphatase inhibitors, TP53 expression enhancer, and apoptosis agonists, which suggests that the elucidated compounds may possess great potential in anticancer activities and inhibiting kinase activity. The reference inhibitor Ulixertinib showed MAPK1 inhibitory potential, which validates the ability of the PASS server to predict.

**Table 3. Biological properties of natural compounds predicted via PASS server.**

| S. No. | Ligand ID | Pa value | Pi value | Biological activity |
|---|---|---|---|---|
| 1. | ZINC02092851 | 0.53 | 0.04 | Anti-inflammatory |
| | | 0.41 | 0.09 | Antineoplastic |
| | | 0.30 | 0.05 | Antineoplastic (breast cancer) |
| 2. | ZINC02130647 | 0.40 | 0.14 | TP53 expression enhancer |
| | | 0.39 | 0.07 | Kinase inhibitor |
| | | 0.38 | 0.08 | Apoptosis agonist |
| 3. | ZINC02133691 | 0.86 | 0.00 | Anti-inflammatory |
| | | 0.57 | 0.05 | Antineoplastic |
| | | 0.39 | 0.00 | Antineoplastic (brain cancer) |
| 4. | Ulixertinib | 0,614 | 0,004 | MAP kinase 1 inhibitor |
| | | 0,291 | 0,023 | Antineoplastic (liver cancer) |
| | | 0,297 | 0,154 | Antineoplastic |

## 3.4. Interaction analysis

Interactions analysis provides important information about inhibitory patterns, bond types, and residual interactions between ligands and the protein. The compounds ZINC02092851, ZINC02130647, and ZINC02133691 were found to interact with the crucial residues of the MAPK1 binding site (**Fig 2**). A detailed binding pattern of ZINC02092851 (yellow), ZINC02130647 (green), and ZINC02133691 (magenta) is illustrated in **Fig 2A**. The **figure** shows that these compounds interact with Gln105, Lys54, and Glu71 (ATP binding sites) of MAPK1, which is crucial for its activity (**Fig 2B**). The structural representation shows that compounds are bound into the deep binding pocket cavity of MAPK1 (**Fig 2C**) [35].

All three compounds exhibited strong interactions with the binding cavity of MAPK1 and its essential residues, including Gln105, Lys54, and Glu71. These residues are essential as they are located within the ATP-binding site of the kinase domain and directly contribute to the functionality of the active site. It is clear from **Fig 3** that all these compounds show common interactions while interacting with the binding residues (ATP-binding) of MAPK1. ATP-

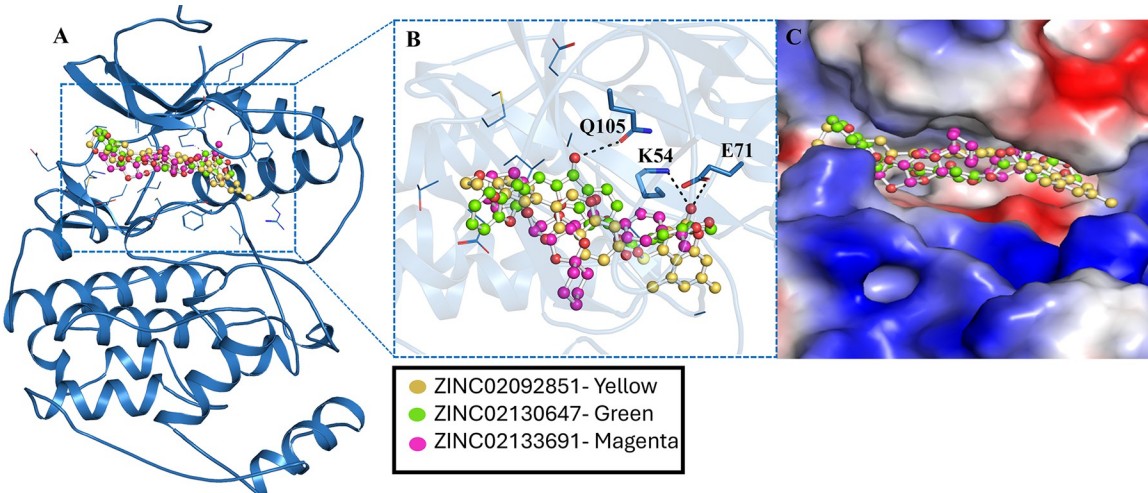

**Fig 2. MAPK1 residual interaction with compounds from ZINC library. (A)** Localization of compounds in MAPK1 binding cavity. **(B)** Magnified interaction. **(C)** Surface potential representation of compounds with MAPK1 binding pocket.

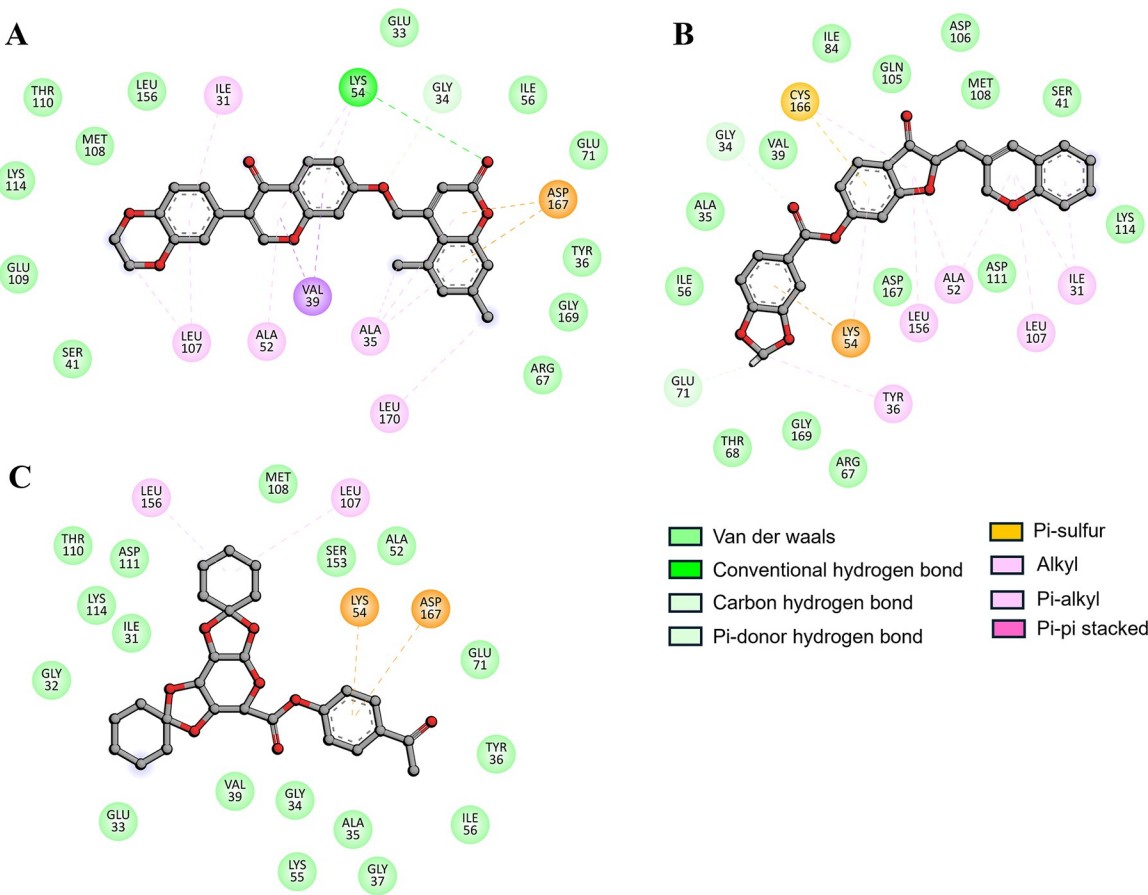

**Fig 3.** Representation of molecular interaction and 2D plots showing detailed interactions of (A) ZINC02092851, (B) ZINC02130647, and (C) ZINC02133691.

binding sites are very important for the kinase activity of the protein. These compounds may inhibit MAPK1 activity by binding to these critical residues [36]. The stable complexes formed between ZINC02092851, ZINC02130647, and ZINC02133691 with MAPK1 alter the protein's structure, impairing its functionality (**Fig 3A–3C**). Notably, these compounds displayed deep localization and strong complementarity within the MAPK1 binding cavity, indicating their potential efficacy in inhibiting MAPK1 binding sites and obstructing ATP accessibility to MAPK1 (**Fig 3**). The strength and specificity of the protein-ligand interactions observed in the study are important for the therapeutic efficacy of the ligand.

## 3.5. Molecular dynamics simulation analysis

After the virtual screening process, we went through the structural dynamics study and stability of MAPK1-ligand complexes through MD simulations. These simulations were carried out on the docked complexes, encompassing three chosen hits from the ZINC library, all within defined solvent conditions. The initial orientations of the selected compounds were employed as starting points for the simulations, which ran for a duration of 200 ns. Throughout the simulations, a range of structural features and parameters were monitored and analyzed. This comprehensive analysis allowed us to gain insights into the dynamic behavior of MAPK1 both before and after interaction with the ligands, shedding light on how these interactions influenced the protein's conformational changes and stability over time [37].

**3.5.1. Structural deviation.** We used the RMSD analysis for MAPK1 and its ligand-bound complexes with ZINC02092851, ZINC02130647, and ZINC02133691 during comprehensive MD simulation. The analysis aimed to assess the structural stability and conformational changes of these molecular systems over a 200 ns simulation period. Here, we found average RMSD values of 0.21 nm, 0.31 nm, 0.24 nm, and 0.26 nm for MAPK1, MAPK1--ZINC02092851, MAPK1-ZINC02130647, and MAPK1-ZINC02133691, respectively (**Table 4**). **Fig 4** displays the structural dynamics of MAPK1 when bound to ZINC02092851, ZINC02130647, and ZINC02133691, which show consistent but small fluctuations. Notably, the MAPK1-ZINC02130647 complex showed more stability with lower RMSD values when compared to the other two complexes after the binding with MAPK1. The RMSD plot in **Fig 4A** depicts that the MAPK1-ZINC02130647 complex has fewer fluctuations compared to the other two complexes [38].

By computing the average fluctuation of all residues, the analysis of RMSF explains the local flexibility and deviation of each residue concerning the mean residual position. The distinctive protein backbone of all four systems, namely MAPK1, MAPK1-ZINC02092851, MAPK1--ZINC02130647, and MAPK1-ZINC02133691, exhibit RMSF values depicting some random peaks in RMSF profiles. Here, we found average RMSF values of 0.08 nm, 0.11 nm, 0.10 nm, and 0.11 nm for MAPK1, MAPK1-ZINC02092851, MAPK1-ZINC02130647, MAPK1--ZINC02133691, respectively. The plot in **Fig 4B** illustrates that the complex MAPK1--ZINC02130647 showed more stability with a lower RMSF value than the other two complexes throughout the trajectory. However, a stable and similar RMSF pattern was illustrated by all three complexes during the entire simulation period [39].

**3.5.2. Structural compactness.** The $R$g provides statistical significance to the protein's secondary structure folding into the tertiary structure and overall functional conformation, which delivers insights into protein stability in a suitable biological system. The $R$g generally assesses the compactness of the protein structure by defining the RMS distance from the collective center of mass of the atoms, for which a decreased $R$g value indicates a compact and stabilized folding during complex formation [40]. The average $R$g values for Free MAPK1, MAPK1-ZINC02092851, MAPK1-ZINC02130647, and MAPK1-ZINC02133691 were found to be 2.16 nm, 2.22 nm, 2.19 nm and 2.19 nm, respectively, showing an appreciable consistency. The time evolution of $R$g is shown in **Fig 5A**, wherein the plot demonstrates that all the complexes appeared stable with consistently robust dynamics and folding, attaining a minimized $R$g. Among all the complexes and the free-state, MAPK1-ZINC02130647 is relatively more stable as it shows more compactness in its structure and has relatively less $R$g value.

SASA is a measure of the surface area of a protein explicitly interacting with its solvent environment through hydrophobic and hydrophilic residual interactions, which divulge the degree of protein folding and compactness [41]. The SASA plot is depicted in **Fig 5B**. The average SASA values for free state MAPK1, MAPK1-ZINC02092851, MAPK1-ZINC02130647, and MAPK1-ZINC02133691 complexes were calculated and quantified, depicting a nearly unaffected SASA during simulations. Here, we found average SASA values of 175.98 nm$^2$, 182.08

**Table 4. The mean values of several MD parameters determined following 200 ns simulations.**

| System | RMSD (nm) | RMSF (nm) | $R$g (nm) | SASA (nm$^2$) | #H-Bonds |
|---|---|---|---|---|---|
| MAPK1 | 0.21 | 0.08 | 2.16 | 175.98 | 252 |
| ZINC02092851 | 0.31 | 0.11 | 2.22 | 182.08 | 247 |
| ZINC02130647 | 0.24 | 0.10 | 2.19 | 178.22 | 247 |
| ZINC02133691 | 0.26 | 0.11 | 2.19 | 180.23 | 257 |

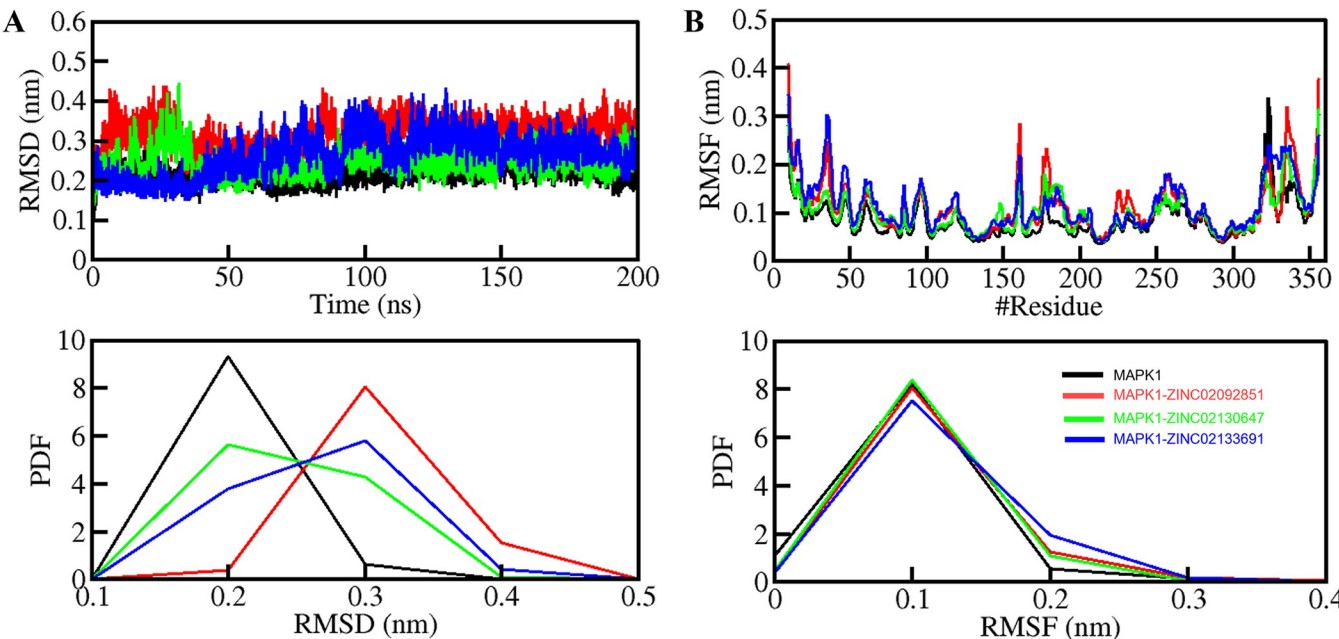

**Fig 4. Structural dynamics of MAPK1 upon ZINC0209285, ZINC02130647, and ZINC02133691.** (A) RMSD plot of MAPK1 complex with ZINC0209285, ZINC02130647, and ZINC02133691. (B) RMSF plot of the MAPK1 and its complex with ZINC0209285, ZINC02130647, and ZINC02133691. The lower panels depict the probability distribution function (PDF) of the values, with the position of the residues indicated by the symbol "#".

$nm^2$, 178.22 $nm^2$, and 180.23 $nm^2$ for MAPK1, MAPK1-ZINC02092851, MAPK1--ZINC02130647, MAPK1-ZINC02133691, respectively (**Table 4**).

**3.5.3. Dynamics of hydrogen bonds.** The hydrogen bonds are a crucial factor in determining the conformational dynamics of a protein [42]. The intramolecular hydrogen bonds were calculated for the MAPK1 free structure and the MAPK1 after binding to

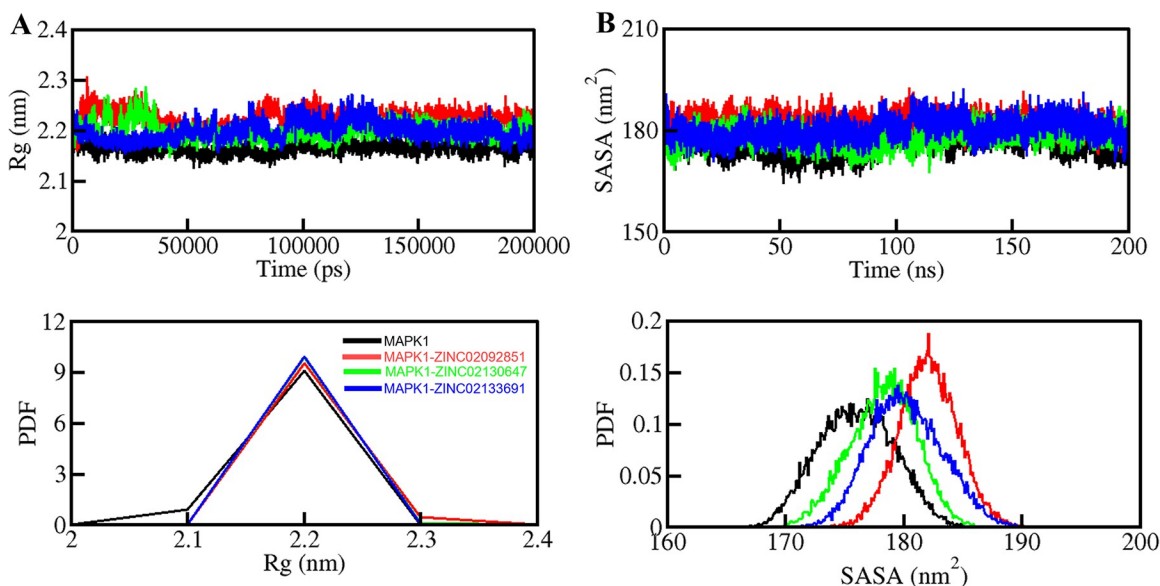

**Fig 5. Structural compactness and folding of MAPK1 with the three selected compounds.** (A) Rg plot and (B) SASA plot of MAPK1 with ZINC02092851, ZINC02130647, and ZINC02133691. Lower panels show the probability distribution function to values as PDF.

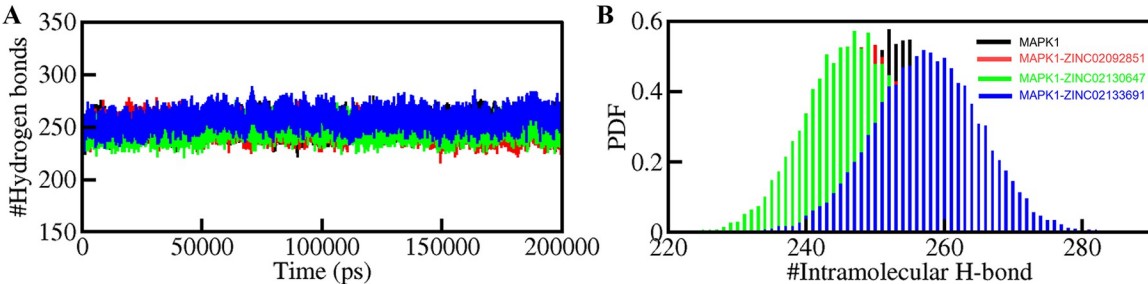

**Fig 6. Analysis of hydrogen bonds.** Time evolution of intramolecular hydrogen bonds (A). The (B) shows the PDF of the hydrogen bond distribution. # represents a number.

ZINC02092851, ZINC02130647, and ZINC02133691. The graphs were plotted throughout 200 ns to evaluate the folding dynamics of the four systems (**Fig 6A**). The plots show a slight change in the number of intramolecular hydrogen bonding interactions between the free protein and the three complexes. The average hydrogen bonds formed before and after ZINC02092851, ZINC02130647, and ZINC02133691 complexes were found to be 252, 247, 247, and 257, respectively (**Table 4**). The PDF for the intramolecular hydrogen bonds was also plotted and it showed good reliability (**Fig 6B**). From the plots, it can be concluded that intramolecular hydrogen bonds in MAPK1 displayed stability throughout the simulation for all four systems [43].

The intermolecular hydrogen bonds formed due to the interaction between the small molecules and the protein were also evaluated. Stable intermolecular hydrogen bonding was observed in all the complexes (**Fig 7**). The MAPK1-ZINC02092851 complex had more hydrogen bonds than the other two complexes and 2–4 hydrogen bonds of the MAPK1-ZINC02092851 complex persist evenly throughout the 200 ns period (**Fig 7A**). The other two complexes, MAPK1-ZINC02130647 and MAPK1-ZINC02133691 showed 1–3 persistent hydrogen bonds in the time evolution (**Fig 7B, 7C**). The findings suggest that the structure of protein-ligand complexes shouldn't change significantly over time. Because of the stable intermolecular hydrogen bonding, the complexes were able to maintain their original docking location throughout time [44].

**3.5.4. Evaluation of secondary structure.** Understanding a protein's conformational behavior and folding mechanism can be accomplished by examining the dynamics of its

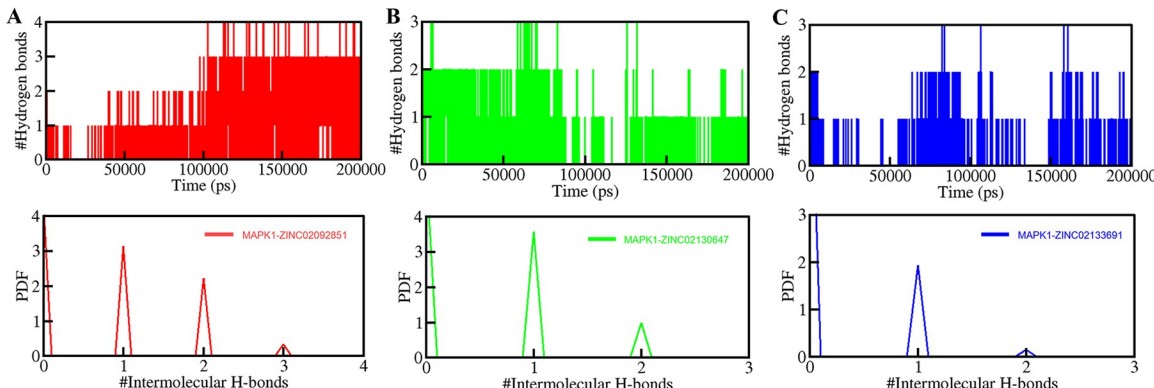

**Fig 7.** Time evaluation and stability of intermolecular hydrogen bonds formed between MAPK1 and (A) ZINC02092851, (B) ZINC02130647, (C) ZINC02133691. The lower panel shows the probability distribution function plot.

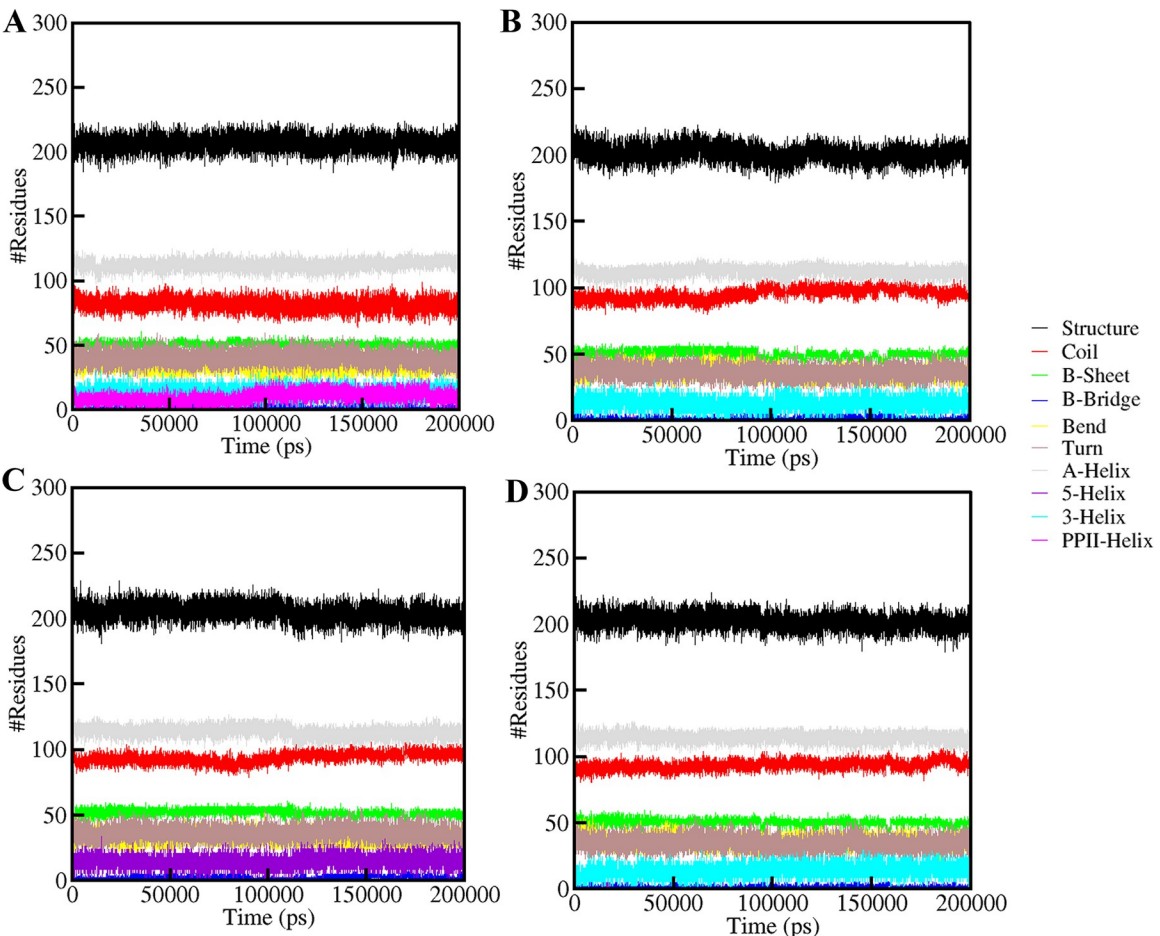

**Fig 8.** The secondary structure content of (A) free MAPK1 (B) MAPK1-ZINC02092851 (C) MAPK1-ZINC02130647, and (D) MAPK1-ZINC02133691.

secondary structure content [45]. We calculated the secondary structural changes in the MAPK1 upon binding of three different compounds, ZINC0209285, ZINC02130647, and ZINC02133691, respectively [46]. The structural components in free MAPK1 remain almost constant and equilibrated throughout the simulation of 200 ns (**Fig 8A**). However, a small change can be seen in the α-helix and β-sheets content of MAPK1 upon compound binding

**Table 5. Average residues participating in secondary structure elements in MAPK1 before and after ligand binding.**

| Element | MAPK1 | MAPK1-ZINC0209285 | MAPK1-ZINC02130647 | MAPK1-ZINC02133691 |
|---|---|---|---|---|
| Coil | 82 | 96 | 94 | 94 |
| β-sheet | 51 | 50 | 52 | 51 |
| β-bridge | 2 | 2 | 2 | 2 |
| Bend | 34 | 36 | 33 | 36 |
| Turn | 40 | 37 | 37 | 35 |
| α-helix | 113 | 113 | 114 | 115 |
| π -helix | 0 | 0 | 0 | 0 |
| $3_{10}$-helix | 14 | 13 | 15 | 14 |
| PPII-Helix | 11 | 0 | 0 | 0 |

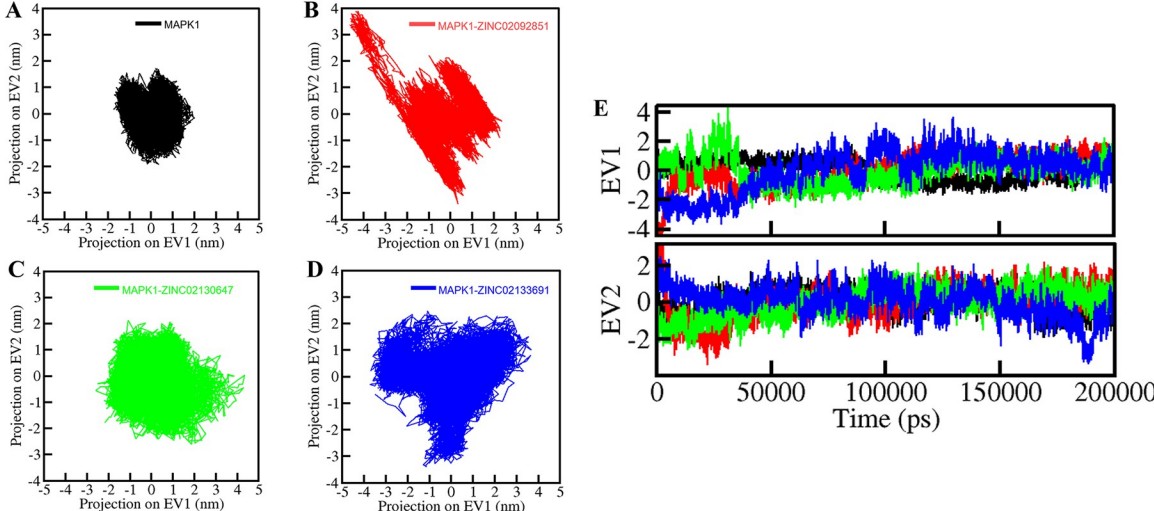

**Fig 9. Principal component analysis.** Two-dimensional conformational projections of (A) MAPK1, (B) MAPK1- ZINC0209285, (C) MAPK1-ZINC02130647, (D) MAPK1-ZINC02133691, (E) Time-dependent projection in Eigen vector-1 and Eigen vector-2.

(**Table 5**). There were minor changes in the average number of residues participating in secondary structure formation in the case of MAPK1-ZINC0209285 (**Fig 8B**), MAPK1--ZINC02130647 (**Fig 8C**), and MAPK1-ZINC02133691 (**Fig 8D**) complexes as compared to the free MAPK1. However, no major change was seen in the secondary structure of MAPK1 upon binding of the selected compounds, which shows strong stability of MAPK1-ZINC0209285, MAPK1-ZINC02130647, and MAPK1-ZINC02133691 complexes. Therefore, employing this structure-based drug design approach to develop selective MAPK1 inhibitors may pave the way for further innovative cancer treatment approaches [47].

### 3.6. Principal component analysis

Principal component analysis (PCA) serves as a valuable tool for assessing conformational changes and collective motions within protein-ligand complexes [36]. In this study, we employed PCA to examine the conformational dynamics of both free MAPK1 and the MAPK1-ZINC02092851, MAPK1-ZINC02130647, and MAPK1-ZINC02133691 complexes. The conformational sampling was conducted by projecting the Cα atoms, as depicted in **Fig 9**. Notably, the essential subspaces occupied by the native form of MAPK1 structure corresponded closely with those of the protein-ligand complexes. None of the complexes extended beyond the eigenvectors (EVs) observed in the free MAPK1 structure. Of particular interest, the MAPK1-ZINC02092851 complex covered a smaller subspace in both EV1 and EV2, which shows a great level of stability within the complex (**Fig 9E**) [37].

### 3.7. Free energy landscape analysis

To gain further insights into the folding mechanisms and energetics of the protein-ligand complexes in solvent conditions, we employed free energy landscape (FEL) analysis [48]. This analysis sheds light on the global and local energy minima achieved by the complexes. **Fig 10** illustrates the FELs of MAPK1 in native form and the MAPK1-ZINC0209285, MAPK1--ZINC02130647, and MAPK1-ZINC02133691 complexes [49]. The color gradients represent the Gibbs free energy (*G*) in kilojoules per mol (kJ/mol). Blue/green regions are the areas of low Gibbs free energy, corresponding to stable states or conformations that the system is most

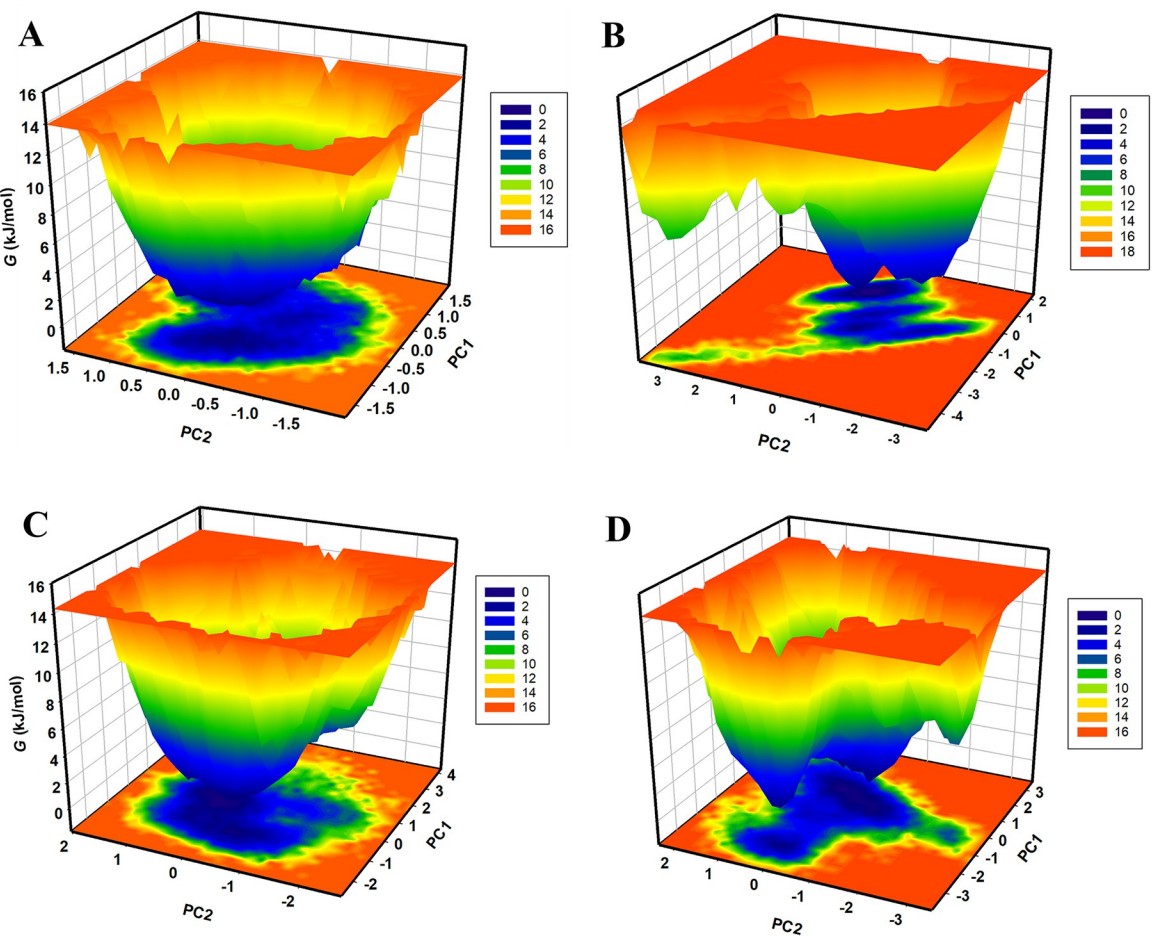

**Fig 10.** The Gibbs free energy landscapes of (A) free MAPK1, (B) MAPK1-ZINC02092851 (C) MAPK1-ZINC02130647, and (D) MAPK1-ZINC02133691.

likely to occupy. These are often interpreted as local or global minima in the free energy landscape. Red regions represent the area of high Gibbs free energy, indicating less favourable or unstable states that the system is less likely to occupy. The native form of MAPK1 shows only one prominent basin, representing a global minimum (**Fig 10A**). Upon binding with ZINC02092851 and ZINC02133691, multiple basins emerged (**Fig 10B–10D**). While in the case of ZINC02130647, a single extensive basin was observed (**Fig 10C**). This suggests that the global minimum of the native MAPK1 was subtly influenced by the binding of the compounds. Overall, the FEL analysis underscores that the binding interactions of ZINC02092851, ZINC02130647, and ZINC02133691 with MAPK1 did not induce the unfolding of the protein throughout the 200 ns simulations. This suggests the structural integrity and stability of the MAPK1-ligand complexes, reinforcing their potential as candidates for further investigation and therapeutic development [50].

Overall, natural compounds have become a focal point in virtual screening for drug discovery due to their diverse chemical structures and bioactive properties [51]. For example, many researchers screen the library of flavonoids targeting different proteins, such as phosphatidylinositol 3-kinase (PI3K), Vascular endothelial growth factor receptor 2 (VEGFR2) and many more, key targets in cancer therapy [52]. The screening process revealed that flavonoids showed strong binding affinities to these targets, suggesting their potential as anticancer

agents. Natural compounds have been increasingly discovered and used for cancer therapy owing to their high molecular diversity, novel functionality, and minimal side effects. These compounds can be utilized as chemopreventive agents because they can efficiently inhibit cell growth, control cell cycle progression, and block several tumour-promoting signalling pathways [53]. Therefore, the compounds identified in this study, ZINC0209285, ZINC02130647, and ZINC02133691, might have high potential to be used as promising candidates as MAPK1 inhibitors for the therapeutic developments against cancer due to their significant binding potential and favorable drug-like properties.

Nonetheless, this study has several limitations that should be considered in future experimental studies. As it relies entirely on computational methods and in silico predictions, the compounds would need to be tested *in vitro* and *in vivo* to confirm their efficacy as MAPK1 inhibitors. Moreover, the study screened only a limited library of about 22,000 natural compounds from the ZINC database, potentially missing other effective inhibitor candidates. The MD simulations were run for a relatively short duration of 200 ns, which may not be sufficient to capture all relevant conformational changes and protein-ligand interactions over longer timescales. Additionally, the study did not assess the selectivity of the identified compounds against other kinases or evaluate potential off-target effects. Despite these limitations, the study provides valuable insights and a foundation for further research into potential MAPK1 inhibitors for cancer therapeutics.

## 4. Conclusions

MAPK1 is an important player in different cancers and acts as a promising target for therapeutic development. A few inhibitors of MAPK1 have been discovered to date, but more potent and specific inhibitors of MAPK1 are required. This study used a thorough molecular docking-based virtual screening method to identify potential inhibitors of MAPK1. After conducting an initial screening of a large natural compound library, 100 possible hits were identified that displayed a notable binding affinity to MAPK1. After doing a more detailed examination, the number of compounds was reduced to 10. These compounds possess desirable natural features and pharmacokinetic characteristics, which makes them highly suitable for the development of drugs. Following the PASS analysis, three compounds were identified as having significant potential for anticancer activity, particularly via targeting MAPK1. Detailed interaction analysis demonstrated that these drugs have strong and stable contacts with key residues within the MAPK1 binding region, implying that they can effectively suppress MAPK1 function. The structural stability and compactness of the MAPK1-ligand complexes were validated by all-atom MD simulations, which showed minimal fluctuations and constant hydrogen bonding interactions. The secondary structure conformation of MAPK1 remained mostly unaltered upon interaction with the chosen compounds, suggesting the stability of these complexes. The essential dynamics analysis through PCA and FEL provided additional evidence of the strength and integrity of the MAPK1-ligand complexes. Taken together, the elucidated compounds, ZINC02092851, ZINC02130647, and ZINC02133691, demonstrate favourable qualities as promising inhibitors of MAPK1. The results presented here establish a solid basis for future research and advancement of these substances as effective remedies in the treatment of cancer.

## Supporting information

**S1 Table. List of filtered docked output\*.**
(DOCX)

**S2 Table. Selected hits and a control molecule with their structures.**
(DOCX)

## Acknowledgments

The authors extend their appreciation to Taif University, Saudi Arabia for supporting this work through project number (TU-DSPP-2024-140). AS is grateful to the Ajman University, UAE for supporting the publication.

## Author Contributions

**Conceptualization:** Md Nayab Sulaimani, Taj Mohammad, Anas Shamsi, Md. Imtaiyaz Hassan.

**Data curation:** Shazia Ahmed, Taj Mohammad, Anas Shamsi, Md. Imtaiyaz Hassan.

**Formal analysis:** Md Nayab Sulaimani, Shazia Ahmed, Farah Anjum, Anas Shamsi, Ravins Dohare.

**Funding acquisition:** Md Nayab Sulaimani, Ravins Dohare.

**Investigation:** Shazia Ahmed, Farah Anjum.

**Methodology:** Ravins Dohare, Md. Imtaiyaz Hassan.

**Project administration:** Ravins Dohare, Md. Imtaiyaz Hassan.

**Resources:** Md. Imtaiyaz Hassan.

**Software:** Md Nayab Sulaimani, Farah Anjum, Taj Mohammad, Ravins Dohare.

**Supervision:** Md Nayab Sulaimani, Taj Mohammad, Anas Shamsi.

**Validation:** Shazia Ahmed, Taj Mohammad.

**Writing – original draft:** Md Nayab Sulaimani, Shazia Ahmed, Taj Mohammad.

**Writing – review & editing:** Farah Anjum, Anas Shamsi, Ravins Dohare, Md. Imtaiyaz Hassan.

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
