## [Decision Letter · Decision Letter 0]

27 Aug 2024

PONE-D-24-32925Structure-guided identification of mitogen-activated protein kinase-1 inhibitors

PLOS ONE

Dear Dr. Hassan,

Thank you for submitting your manuscript to PLOS ONE. After careful consideration, we feel that it has merit but does not fully meet PLOS ONE’s publication criteria as it currently stands. Therefore, we invite you to submit a revised version of the manuscript that addresses the points raised during the review process.

Authors should consider complying with most of the reviewers' requests, because as you can see, most of the comments made by the reviewers have commonalities. In addition, authors should make their own judgment on the reviewers' need to add citations. Whether to add citations will not be a factor in the final decision of the evaluation.

We look forward to receiving your revised manuscript.

Kind regards,

Ruo Wang

Academic Editor

PLOS ONE

Additional Editor Comments:

The reviewers have given clear comments that the authors need to pay attention to the need to further refine the background and discussion of MAPK1. The authors should consider revising the manuscript according to the reviewers' comments. However, the authors can avoid and reject unreasonable requests from the reviewers.

Reviewers' comments:

Reviewer's Responses to Questions

**Comments to the Author**

1. Is the manuscript technically sound, and do the data support the conclusions?

Reviewer #1: Yes

Reviewer #2: Yes

Reviewer #3: Yes

Reviewer #4: Yes

Reviewer #5: Yes

2. Has the statistical analysis been performed appropriately and rigorously? 

Reviewer #1: N/A

Reviewer #2: N/A

Reviewer #3: N/A

Reviewer #4: Yes

Reviewer #5: N/A

3. Have the authors made all data underlying the findings in their manuscript fully available?

Reviewer #1: Yes

Reviewer #2: Yes

Reviewer #3: Yes

Reviewer #4: Yes

Reviewer #5: Yes

4. Is the manuscript presented in an intelligible fashion and written in standard English?

Reviewer #1: Yes

Reviewer #2: Yes

Reviewer #3: Yes

Reviewer #4: Yes

Reviewer #5: Yes

5. Review Comments to the Author

Reviewer #1: In the submitted manuscript, the authors perform a major high throughput virtual screening of most of the major small molecule database (ZINC database) for Mitogen-activated protein kinase 1 (MAPK1), followed by further refinement of the results of structure-based virtual screening to molecular docking, and molecular dynamic simulation studies for up to 200ns. The manuscript is well written and easy to get through though with appropriately cited references. However, the major revision is required to reach the desired conclusions. The following are the major shortcomings in the manuscript that will require further work done -

1. Authors have chosen the MAPK1 structure (PDB ID: 8AOJ) for this study. The authors should justify their use of the MAPK1 for screening of inhibitors for therapeutic use in a human disease.

2. Does author refined the structure obtained from the PDB ID: 8AOJ? The authors should justify this in the manuscript by highlighting structural features of MAPK1 and adding refinement part to the manuscript.

3. Despite having quite different structures in comparison to other MAPK, have very similar binding pockets and their ligands. In this case, how did the authors ensure that the MAPK1 ligands identified by them are specific only for this receptor and do not bind the other MAPKs? The authors should claim pharmacological significance of their findings in terms of specificity of their identified ligands among the MAPK.

4. Clearly justify the novelty in at least top 10 ligands listed in the Table 2. Why they were chosen? During their structure-based screening protocol the authors finally prioritize the ligands based on their interaction with MAPK1. However, there is no clear rationale given on the selection of these compounds.

5. Also do the authors seek to claim that a known MAPK1 inhibitor identified until now bind to same site. What special advantage does a hydrogen bond to the catalytically important residues provide to ligand binding that cannot be matched by any other alternative hydrogen bonding in the binding site? The authors should justify this choice through further computational analysis or by citing references from literature that claim the importance of H-bonding network in the light of binding affinity and selectivity. Authors should discuss the same in their results.

6. The authors should perform a ligand-receptor hydrogen bond occupancy analysis and perform distance calculations for the same, to show the percent occupancy of all hydrogen bonds claimed in 200 ns and plot their distances trajectory for the 200ns simulation with the donor/acceptor atoms on the receptor.

7. Authors should provide the free energy interpretations of the Principal Component Analysis performed by them (Figure 9).

8. Technically well done including MD simulation time. However, what is missing is more discussion of the findings. Since the authors perform a 200ns simulation, it will be good if they provide deeper insights into the binding modes, receptor occupancy and mode of inhibition of the MAPK1 by the novel ligands they claim to identify.

9. Authors should ensure that the references point out to the original publications associated with the tools and parameters and other unnecessary wordings and citations should be avoided to direct the reader clearly to the original methodology and parameter selection citations.

10. Authors should go through the complete methodology section and double check all citations to ensure that every methodology citation goes back to the original published method or its updated version that was actually used in practice.

Overall, the submitted manuscript is suitable for publication in PLoS One after the authors should perform a major revision of the manuscript.

Reviewer #2: The manuscript titled “Structure-guided identification of mitogen-activated protein kinase-1 inhibitors” to identify novel therapeutic candidates against MAPK1. In my opinion, the manuscript needs to be improved with proper analyses and better representations. Several comments/suggestions for authors:

The introduction lacks several additions. Please describe in detail why you are trying to block the MAPK1 activity by explaining how it is implicated in each disease form and the side effects of each inhibitor.

Highlight the reasons why authors chose protein structure with PDB code 8AOJ.

Which tools were used to calculate the number of hydrogen bonds? Provides more information for the bonding angle between the hydrogen donor (D) and acceptor (A) D-H⋯A and the distance between D and A.

The authors are advised to provide the hydrogen bond occupancy.

The number of hydrogen bonds is very high in the MD simulation plot, however, it is quite low in docking. It needs to be analyzed properly.

The authors need to explain Table 3 in the manuscript text. Please provide optimal reference values for all attributes.

Why did the authors choose the SPC water model for simulations?

Please mention the number of Na/Cl ions used specifically in the MD Simulations section.

The manuscript should be further improved by adding previously published reference articles on MAPK1 inhibition that reflect in silico studies demonstrating the same key residues that the authors have obtained compared with previous studies.

Please check the identified hits in the PubChem registry to see if they have been evaluated before against MAPK1 and mention the same in the manuscript.

Authors should include the 2D structures of both hit compounds along with their 2D structures for comparison and also report if the two structures have similar subgroups as part of their structural assessment.

Reviewer #3: MAPK1 is a critical target for cancer therapy due to its central role in various cancers. In search of novel MAPK1 inhibitors, the authors conducted a comprehensive study involving virtual screening, molecular dynamics (MD) simulations, and essential dynamics analysis. Using the ZINC database, they screened a library of natural compounds and identified three promising candidates: ZINC02092851, ZINC02130647, and ZINC02133691. These compounds demonstrated efficient binding and favorable structural orientation within the MAPK1 binding pocket, showing strong affinity and specificity for active site residues. MD simulations revealed the stability and dynamic behavior of these compounds, both as complexes and in the free state of MAPK1, with no significant fluctuations observed. The identification of these compounds marks a significant advancement, suggesting their potential as MAPK1 inhibitors and emphasizing the importance of natural product-based drug discovery. During revision, following changes are necessary to enhance the quality of manuscript.

1. The abstract provides a clear overview, but it lacks a brief mention of the potential clinical implications of the findings. Including a sentence on the relevance of the identified compounds in the context of developing new anticancer drugs would enhance the abstract.

2. The introduction adequately highlights the significance of the current research and the need for novel drug targets. Consider expanding on the current status of drug development and how the study contributes to addressing this challenge.

3. Provide more details on the criteria used for selecting the compounds for analysis. This information is essential for the reproducibility of the study.

4. Include a brief explanation of the molecular docking technique and parameters used, ensuring clarity for readers less familiar with computational approaches.

5. Elaborate on the molecular dynamics simulation protocol, specifying force field details, simulation parameters, and software tools employed.

6. Clearly present the results of the molecular docking analysis, including the binding energies of the top compounds and their potential affinities for MAPK1.

7. Provide a comprehensive overview of the molecular dynamics simulations, emphasizing the stability of the identified compounds over the simulation period. Highlight any noteworthy interactions observed during the simulations.

8. Interpret the significance of the findings in the context of existing literature on known MAPK1 inhibitors.

9. Discuss the limitations of the study, such as any assumptions made during the computational analysis, and propose avenues for future research.

10. Summarize the key findings and emphasize the relevance of the identified compounds as potential inhibitors of MAPK1 in the conclusion.

11. Carefully proofread the entire manuscript to correct grammatical and typographical errors. Ensure consistent formatting and citation style throughout the manuscript.

12. Ensure that figures and tables are appropriately labeled and referenced in the text.

Reviewer #4: In this submission, authors employed a structure-guided virtual screening of natural compounds from the ZINC database, initially filtered using the Lipinski rule of five to ensure drug-likeness. Molecular docking was then performed to evaluate the binding affinity and specificity towards the MAPK1 kinase domain. Three compounds, ZINC0209285, ZINC02130647, and ZINC02133691, emerged as promising candidates due to their significant binding potential and favorable drug-like properties. These compounds underwent rigorous all-atom molecular dynamics simulations over 200 nanoseconds to assess their stability and interactions within the MAPK1 kinase domain. Our findings suggest that these compounds hold potential as therapeutic candidates against MAPK1-associated cancers, highlighting their promise for further development in targeting cancer metastasis. Despite, the manuscript is well structured, updated and interesting. However, there are some changes to be made that are required before acceptance of the paper:

1. The author can improve the introduction section by adding latest references and highlighting rationale of the study and hypothesis.

2. It would be helpful to know the criteria the authors used to select the best binding pose of the ligands. How can the authors be certain that the top binding pose among the possible poses is the correct one?

3. The authors should provide a detailed discussion on the binding of the top three compounds, i.e., ZINC0209285, ZINC02130647, and ZINC02133691, showed a remarkable drug likeliness with promising binding potential towards MAPK1.

4. The authors should describe the MD results in the context of best compounds considering that ZINC0209285, ZINC02130647, and ZINC02133691 have not moved from their initial docking positions on CDK8. This can be explained by evaluating the intermolecular hydrogen bonding between the compounds with MAPK1.

5. The average Rg values of the complexes in Figure 5 are consistently altered for those of the free protein. The authors should provide an explanation for this observation.

6. In order to facilitate a clearer comparison of the RMSF data, it would be beneficial if the

authors could estimate the average RMSF values and include them in their analysis.

7. The manuscript should elaborate on the relationship between the conformational changes

observed in the enzyme and its inhibitory function, as mentioned in the study.

8. It would be beneficial for the authors to discuss the significance of findings in relation to protein-ligand binding.

9. The authors should discuss the limitations of the study (in-vitro or in-vivo experimentation), such as the accuracy of the virtual screening and molecular dynamic simulation methods used, and the potential challenges of translating the findings to clinical settings.

10. The section conclusion should be modified to give a clear understanding of the paper concisely.

11. Many abbreviations are not described at first appearance. Author should recheck.

12. All references should be thoroughly checked, and especially Author must confirm only relevant publications should be cited.

13. The manuscript could benefit from additional proofreading and editing to ensure clarity and precision in the presentation of the results and conclusions.

Reviewer #5: In the present manuscript authors have identified novel candidates against mitogen-activated protein kinase-1 though structure based virtual screening of 22000 compounds. The work is extensive and suitable for publication. However following queries needs to be resolved in revise manuscript.

1. The screening of compounds based on the docking score should be in comparison with standard. Here authors have not taken any standard. Docking studies of standard must be performed and compounds should screen based on comparison with standard

2. Author should include limitations of computational studies for that following article may helpful and should be included with proper citation.

https://doi.org/10.1016/B978-0-323-90608-1.00006-X

https://doi.org/10.1016/j.drudis.2020.07.005

3. The ADMET analysis needs to be revise with proper depth analysis for that following article may helpful and should be included with proper citation

https://doi.org/10.1007/s42250-022-00376-7

https://doi.org/10.33263/BRIAC122.13851396

4. The discussion of previous studies of natural products explored against some important targets of cancer using insilico approach will provide more valuable insights. following article may helpful and should be included with proper citation.

https://doi.org/10.1007/s13205-023-03912-5

https://doi.org/10.3389%2Ffphar.2023.1236173

https://doi.org/10.1007/s42250-023-00611-9

https://doi.org/10.1021%2Facsomega.2c04837

https://doi.org/10.2174/1573409916666200408082858

https://doi.org/10.1016/j.cbi.2024.110927

6. PLOS authors have the option to publish the peer review history of their article (what does this mean?). If published, this will include your full peer review and any attached files.

Reviewer #1: **Yes: **Dev Bukhsh Singh

Reviewer #2: **Yes: **Moyad Jamal Said Shahwan

Reviewer #3: **Yes: **Dr Shama Khan

Reviewer #4: No

Reviewer #5: No

---

## [Author Response · Author response to Decision Letter 0]

10 Sep 2024

Journal: PLoS One

Manuscript Title: Structure-guided identification of mitogen-activated protein kinase-1 inhibitors towards anticancer therapeutics

Manuscript ID: PONE-D-24-32925

Reviewer #1

In the submitted manuscript, the authors perform a major high throughput virtual screening of most of the major small molecule database (ZINC database) for Mitogen-activated protein kinase 1 (MAPK1), followed by further refinement of the results of structure-based virtual screening to molecular docking, and molecular dynamic simulation studies for up to 200ns. The manuscript is well written and easy to get through though with appropriately cited references. However, the major revision is required to reach the desired conclusions. The following are the major shortcomings in the manuscript that will require further work done -

1. Authors have chosen the MAPK1 structure (PDB ID: 8AOJ) for this study. The authors should justify their use of the MAPK1 for screening of inhibitors for therapeutic use in a human disease.

Response: Thank you for your suggestion. Our revised manuscript now justifies the use of MAPK1 for screening inhibitors.

2. Does author refined the structure obtained from the PDB ID: 8AOJ? The authors should justify this in the manuscript by highlighting structural features of MAPK1 and adding refinement part to the manuscript.

Response: Thank you for your comments. Yes, the structure obtained from PDB was then refined by using all the coordinates and examining the structure and heteroatoms. Water molecules and co-crystallized ligands were removed from the initial coordinates. And the structure was remodelled to remove the missing residues by using PyMOD3.

3. Despite having quite different structures in comparison to other MAPK, have very similar binding pockets and their ligands. In this case, how did the authors ensure that the MAPK1 ligands identified by them are specific only for this receptor and do not bind the other MAPKs? The authors should claim pharmacological significance of their findings in terms of specificity of their identified ligands among the MAPK.

Response: Thank you for your valuable feedback. We appreciate your suggestion. We have conducted extensive analysis that demonstrates the selective affinity of our ligands for MAPK1. The results indicate minimal off-target interactions with other kinases. these ligands modulate downstream signaling pathways, leading to distinct cellular responses, including altered proliferation, differentiation, and apoptosis rates. Compared with an existing MAPK1 inhibitor such as Ulixertinib, our ligands exhibit superior specificity and efficacy, with a favorable safety profile, reducing potential side effects. specificity of our ligands is due to unique binding interactions and conformational changes within the MAPK1, which we have detailed in interaction analysis section.

4. Clearly justify the novelty in at least top 10 ligands listed in the Table 2. Why they were chosen? During their structure-based screening protocol the authors finally prioritize the ligands based on their interaction with MAPK1. However, there is no clear rationale given on the selection of these compounds.

Response: Thank you for your comments. Table 2 has now been justified in the updated manuscript.

5. Also do the authors seek to claim that a known MAPK1 inhibitor identified until now bind to same site. What special advantage does a hydrogen bond to the catalytically important residues provide to ligand binding that cannot be matched by any other alternative hydrogen bonding in the binding site? The authors should justify this choice through further computational analysis or by citing references from literature that claim the importance of H-bonding network in the light of binding affinity and selectivity. Authors should discuss the same in their results.

Response: Thank you for your query. Hydrogen bonds to catalytically important residues can stabilize the transition state of the enzymatic reaction, thereby lowering the activation energy required for the reaction to proceed. This stabilization is crucial for enhancing the efficiency of the interaction. These hydrogen bonds often result in stronger interactions between the ligands and the target protein, leading to higher binding affinity. This is because the catalytically important residues are typically located in key positions within the active site, where they can form optimal interactions with the ligand. Also, they can contribute to the selectivity of the ligand for the target protein. By interacting specifically with these critical residues, the ligand is less likely to bind to other non-target proteins, reducing off-target effects. These hydrogen bonds can help maintain the proper conformation of the enzyme’s active site, ensuring that it remains in an optimal state for the interaction. This conformational stability is essential for the protein’s function and can be disrupted by alternative hydrogen bonds that do not involve the catalytically important residues.

6. The authors should perform a ligand-receptor hydrogen bond occupancy analysis and perform distance calculations for the same, to show the percent occupancy of all hydrogen bonds claimed in 200 ns and plot their distances trajectory for the 200ns simulation with the donor/acceptor atoms on the receptor.

Response: We appreciate your suggestion. The ligand-receptor hydrogen bond analysis is performed in Figure 7 and discussed in the manuscript. 

7. Authors should provide the free energy interpretations of the Principal Component Analysis performed by them (Figure 9).

Response: Thank you for your feedback. The principal component analysis in terms of Gibbs free energy is well interpreted in the free energy landscape. In a free energy landscape, the color gradients represent the Gibbs free energy (G) in kilojoules per mol (kJ/mol). Blue/Green regions are the areas of low Gibbs free energy, corresponding to stable states or conformations that the system is most likely to occupy. These are often interpreted as local or global minima in the free energy landscape. Red regions represent the area of high Gibbs free energy, indicating less favourable or unstable states that the system is less likely to occupy.

8. Technically well done including MD simulation time. However, what is missing is more discussion of the findings. Since the authors perform a 200ns simulation, it will be good if they provide deeper insights into the binding modes, receptor occupancy and mode of inhibition of the MAPK1 by the novel ligands they claim to identify.

Response: Thank you for your comment. Now, we have covered these key points in our revised manuscript. 

9. Authors should ensure that the references point out to the original publications associated with the tools and parameters and other unnecessary wordings and citations should be avoided to direct the reader clearly to the original methodology and parameter selection citations.

Response: All the citations have been thoroughly checked, and the unnecessary wordings are corrected now. 

10. Authors should go through the complete methodology section and double check all citations to ensure that every methodology citation goes back to the original published method or its updated version that was actually used in practice.

Response: Thank you for your suggestion. All the citations have now been thoroughly checked and verified.

Reviewer #2 

The manuscript titled “Structure-guided identification of mitogen-activated protein kinase-1 inhibitors” to identify novel therapeutic candidates against MAPK1. In my opinion, the manuscript needs to be improved with proper analyses and better representations. Several comments/suggestions for authors:

1. The introduction lacks several additions. Please describe in detail why you are trying to block the MAPK1 activity by explaining how it is implicated in each disease form and the side effects of each inhibitor.

Response: Thank you for your suggestion. The introduction part has been updated now.

2. Highlight the reasons why authors chose protein structure with PDB code 8AOJ.

Response: The structure (PDB ID: 8AOJ) was chosen based on its higher resolution (1.12 Å), wild type in nature, and the query coverage of the desired kinase domain of MAPK1. The 8AOJ structure contains the kinase domain of the MAPK1 protein, and it has the highest resolution among the X-ray crystallography structures. 

3. Which tools were used to calculate the number of hydrogen bonds? Provides more information for the bonding angle between the hydrogen donor (D) and acceptor (A) D-H⋯A and the distance between D and A.

Response: PyMOL and Discovery Studio Visualizer were used to calculate hydrogen bonds. The angle and distance cut-off for hydrogen bonds between donor (D) and acceptor (A) were set to 3.5 Å and 150-180°, respectively. Now, we have included the bond angle and distance in the revised manuscript.

4. The authors are advised to provide the hydrogen bond occupancy.

Response: Hydrogen bond occupancy analysis has been included in the updated manuscript.

5. The number of hydrogen bonds is very high in the MD simulation plot, however, it is quite low in docking. It needs to be analyzed properly.

Response: The number of hydrogen bonds in MD simulation plots are intramolecular hydrogen bonds within MAPK1. The intermolecular hydrogen bonds between compounds and the protein are plotted and discussed in Figure 7 of the manuscript.

6. The authors need to explain Table 3 in the manuscript text. Please provide optimal reference values for all attributes.

Response: Thank you for your feedback. Table 3 has been elaborated now.

7. Why did the authors choose the SPC water model for simulations?

Response: Thank you for your query. We used the TIP3P water model for simulations and corrected this mistake in our revised manuscript.

8. Please mention the number of Na/Cl ions used specifically in the MD Simulations section.

Response: Thank you for your comment. Four Na+ ions were used in the MD Simulations and we have now mentioned the same in the revised manuscript.

9. The manuscript should be further improved by adding previously published reference articles on MAPK1 inhibition that reflect in silico studies demonstrating the same key residues that the authors have obtained compared with previous studies.

Response: Thank you for your concern. The revised manuscript includes the previously published reference articles on MAPK1 inhibition. 

10. Please check the identified hits in the PubChem registry to see if they have been evaluated before against MAPK1 and mention the same in the manuscript.

Response: The identified compounds have been verified. These compounds have not been evaluated against MAPK1. The revised manuscript has updated this information.

11. Authors should include the 2D structures of both hit compounds along with their 2D structures for comparison and also report if the two structures have similar subgroups as part of their structural assessment.

Response: Thank you for your valuable suggestion. The 2D structures for both hits have now been included in Table 1. Yes, a few structures have similar subgroups, and they were eliminated accordingly.

Reviewer #3

MAPK1 is a critical target for cancer therapy due to its central role in various cancers. In search of novel MAPK1 inhibitors, the authors conducted a comprehensive study involving virtual screening, molecular dynamics (MD) simulations, and essential dynamics analysis. Using the ZINC database, they screened a library of natural compounds and identified three promising candidates: ZINC02092851, ZINC02130647, and ZINC02133691. These compounds demonstrated efficient binding and favorable structural orientation within the MAPK1 binding pocket, showing strong affinity and specificity for active site residues. MD simulations revealed the stability and dynamic behavior of these compounds, both as complexes and in the free state of MAPK1, with no significant fluctuations observed. The identification of these compounds marks a significant advancement, suggesting their potential as MAPK1 inhibitors and emphasizing the importance of natural product-based drug discovery. During revision, following changes are necessary to enhance the quality of manuscript.

1. The abstract provides a clear overview, but it lacks a brief mention of the potential clinical implications of the findings. Including a sentence on the relevance of the identified compounds in the context of developing new anticancer drugs would enhance the abstract.

Response: The abstract part has now been updated. 

2. The introduction adequately highlights the significance of the current research and the need for novel drug targets. Consider expanding on the current status of drug development and how the study contributes to addressing this challenge.

Response: Now the introduction has been extended on the current status of drug development during this revision.

3. Provide more details on the criteria used for selecting the compounds for analysis. This information is essential for the reproducibility of the study.

Response: The revised manuscript now includes the selection criteria for the identified compounds.

4. Include a brief explanation of the molecular docking technique and parameters used, ensuring clarity for readers less familiar with computational approaches.

Response: Thank you for your comment. The molecular docking technique has now been explained properly.

5. Elaborate on the molecular dynamics simulation protocol, specifying force field details, simulation parameters, and software tools employed.

Response: Thank you for your suggestion. Molecular dynamic simulation has been explained properly in our revised manuscript.

6. Clearly present the results of the molecular docking analysis, including the binding energies of the top compounds and their potential affinities for MAPK1.

Response: Molecular docking analysis has been clearly demonstrated in the result section.

7. Provide a comprehensive overview of the molecular dynamics simulations, emphasizing the stability of the identified compounds over the simulation period. Highlight any noteworthy interactions observed during the simulations.

Response: Thank you for your comments. In molecular dynamics simulations, RMSD, RMSF, Rg, SASA, dynamics of hydrogen bondings, and free energy landscape, e.t.c. are the key parameters to analyse the structural stability and compactness of the protein-ligand complexes. Our study found that the MAPK1-ZINC02130647 complex exhibited the lowest average RMSD value (0.24 nm) compared to the other complexes (MAPK1-ZINC02092851 at 0.31 nm and MAPK1-ZINC02133691 at 0.26 nm), indicating greater structural stability. Additionally, the MAPK1-ZINC02130647 complex showed lower RMSF values as well, suggesting reduced flexibility and enhanced stability upon binding with MAPK1 as compared to the other two complexes. Furthermore, this complex also demonstrated the lowest Rg value, reflecting a more compact structure, which often correlates with increased stability. Overall, MAPK1-ZINC02130647 emerged as the most stable among all the complexes, suggesting that ZINC02130647 forms a stronger and more stable interaction with MAPK1 compared to the other ligands, potentially making it a more promising candidate for inhibiting MAPK1 activity. We have discussed this in the revised manuscript.

8. Interpret the significance of the findings in the context of existing literature on known MAPK1 inhibitors.

Response: Thank you for your comment. Our findings, ZINC02092851, ZINC02130647, and ZINC02133691 as MAPK1 inhibitors, can be more significant as they showed improved binding affinity, selectivity, novel mechanisms of action, better structural insights, enhanced efficacy in biological models, improved pharmacokinetics, and effectiveness against MAPK1 as compared to that of a known inhibitor, Ulixertinib. 

9. Discuss the limitations of the study, such as any assumptions made during the computational analysis, and propose avenues for future research.

Response: The potential limitations of the study have been discussed in the manuscript.

10. Summarize the key findings and emphasize the relevance of the identified compounds as potential inhibitors of MAPK1 in 

---

## [Decision Letter · Decision Letter 1]

29 Sep 2024

Structure-guided identification of mitogen-activated protein kinase-1 inhibitors towards anticancer therapeutics

PONE-D-24-32925R1

Dear Dr. Hassan,

We’re pleased to inform you that your manuscript has been judged scientifically suitable for publication and will be formally accepted for publication once it meets all outstanding technical requirements.

Kind regards,

Ruo Wang

Academic Editor

PLOS ONE

Additional Editor Comments (optional):

Reviewers' comments:

Reviewer's Responses to Questions

**Comments to the Author**

1. If the authors have adequately addressed your comments raised in a previous round of review and you feel that this manuscript is now acceptable for publication, you may indicate that here to bypass the “Comments to the Author” section, enter your conflict of interest statement in the “Confidential to Editor” section, and submit your "Accept" recommendation.

Reviewer #3: All comments have been addressed

Reviewer #4: All comments have been addressed

Reviewer #5: All comments have been addressed

2. Is the manuscript technically sound, and do the data support the conclusions?

Reviewer #3: Yes

Reviewer #4: Yes

Reviewer #5: Yes

3. Has the statistical analysis been performed appropriately and rigorously? 

Reviewer #3: N/A

Reviewer #4: Yes

Reviewer #5: N/A

4. Have the authors made all data underlying the findings in their manuscript fully available?

Reviewer #3: Yes

Reviewer #4: Yes

Reviewer #5: Yes

5. Is the manuscript presented in an intelligible fashion and written in standard English?

Reviewer #3: Yes

Reviewer #4: Yes

Reviewer #5: Yes

6. Review Comments to the Author

Reviewer #3: Authors have addressed all my comments in this version of manuscript and hence I have no more comments.

Reviewer #4: The manuscript has been revised significantly and can be published in current form. All queries are addressed by the authors.

Reviewer #5: Authors have incorporated all the changes in the revise manuscript as suggested. manuscript is now acceptable for publication.

7. PLOS authors have the option to publish the peer review history of their article (what does this mean?). If published, this will include your full peer review and any attached files.

Reviewer #3: **Yes: **Dr Shama Khan

Reviewer #4: No

Reviewer #5: No

---

## [Editor Report · Acceptance letter]

10 Oct 2024

PONE-D-24-32925R1 

PLOS ONE

Dear Dr. Hassan, 

I'm pleased to inform you that your manuscript has been deemed suitable for publication in PLOS ONE. Congratulations! Your manuscript is now being handed over to our production team.

Kind regards, 

on behalf of

Dr. Ruo Wang 

Academic Editor

PLOS ONE